# Liver Fibrosis Leading to Cirrhosis: Basic Mechanisms and Clinical Perspectives

**DOI:** 10.3390/biomedicines12102229

**Published:** 2024-09-30

**Authors:** Kaumudi Somnay, Priyanka Wadgaonkar, Nidhishri Sridhar, Prarath Roshni, Nachiketh Rao, Raj Wadgaonkar

**Affiliations:** 1New York Presbyterian Hospital, Queens, New York, NY 11355, USA; 2New York Digestive Disease Center, Queens, New York, NY 11355, USA; 3Montefiore Medical Center, Bronx, New York, NY 10461, USA; 4SUNY Downstate Medical Center, Brooklyn, New York, NY 11203, USA; raj.wadgaonkar@downstate.edu

**Keywords:** cirrhosis, liver fibrosis, hepatic stellate cells, reversibility

## Abstract

Liver fibrosis is the pathological deposition of extracellular matrix rich in fibrillar collagen within the hepatocytes in response to chronic liver injury due to various causes. As the condition advances, it can progress to cirrhosis, the late stages of which are irreversible. Multiple pathophysiological mechanisms and cell types are responsible for the progression of liver fibrosis and cirrhosis. Hepatic stellate cells and myofibroblast activation represent a key event in fibrosis. Capillarization of liver sinusoidal endothelial cells further contributes to extracellular matrix deposition and an increase in portal pressure. Macrophages and neutrophils produce inflammatory cytokines and participate in activating hepatic stellate cells. Although initially believed to be irreversible, early stages of fibrosis are now found to be reversible. Furthermore, advances in noninvasive imaging and serum studies have changed and improved how cirrhosis can be evaluated and monitored. Although there are currently no specific approved therapies to reverse liver fibrosis, management of underlying diseases has been found to halt the progression, and to an extent, even reverse liver fibrosis, preventing further liver injury and cirrhosis-related complications.

## 1. Introduction

Liver fibrosis progression is described through stages F1 to F4 or cirrhosis. Cirrhosis is characterized by distorted hepatic architecture and function with the formation of regenerative nodules [1]. Any chronic liver condition can result in hepatic fibrosis, which is defined by the progressive formation of scar tissue as a result of excessive extracellular matrix protein deposition. Untreated, cirrhosis leads to serious health problems, including esophageal varices, ascites, hepatic encephalopathy, and hepatocellular carcinoma [2]. Although cirrhosis has been conventionally defined as irreversible liver fibrosis, recent evidence suggests that even cirrhosis is reversible in the early stages. The exact threshold at which fibrosis becomes irreversible remains unclear [3]. There are currently no available treatment options approved that specifically reverse fibrosis in patients with advanced liver disease. However, there is now evidence that fibrosis can be reversed once the underlying pathology is treated [4,5,6]. This review focuses on updates in the pathophysiology, evaluation, and management of cirrhosis.

### 1.1. Etiology

The most prevalent causes of cirrhosis in developed nations are non-alcoholic fatty liver disease (10%), chronic viral hepatitis B or C (10%), and alcoholic liver disease (60–70%) [6]. Other, less common, causes include:⬤Autoimmune: autoimmune hepatitis, primary sclerosing cholangitis, primary biliary cholangitis;⬤Genetic: hemochromatosis, Wilson disease, alpha-1 antitrypsin deficiency, polycystic liver disease;⬤Medications: methotrexate, isoniazid, phenytoin, amiodarone;⬤Post-hepatic: right-sided heart failure, Budd–Chiari syndrome, biliary obstruction from biliary atresia, congenital biliary cysts, or cystic fibrosis;⬤Infection: syphilis, echinococcosis;⬤Idiopathic or cryptogenic cirrhosis.

### 1.2. Epidemiology

Cirrhosis caused more than 1.32 million (2.4%) deaths globally in 2017, an upward trend from less than 899,000 deaths due to cirrhosis in 1990 [7]. The CDC’s 2022 National Health Interview Survey mortality data in the United States show 54,803 recorded deaths, with a rate of 16.4 deaths per 100,000 population. The 2018 survey indicates that approximately 4.5 million adults aged 18 and older have been diagnosed with liver disease, accounting for 1.8% of this demographic. Prevalence is higher in males, older age, those who meet the criteria for poverty, lower educational levels, black race, and Mexican Americans [8].

### 1.3. Pathophysiology

The progression towards cirrhosis involves many pathophysiologic mechanisms including liver inflammation, fibrogenesis, and angiogenesis. The combination of these processes leads to changes in the microvascular circulation of the liver, decreasing perfusion to hepatic endothelial cells ultimately causing their dysfunction. Hepatic endothelial dysfunction is the primary driver of increased hepatic portal pressure, the main mechanism for many of the complications seen in cirrhosis [9].

The origin of fibrogenesis involves the interplay of various cell types (Figure 1):⬤Hepatic stellate cells (HSCs) are considered the primary source of extracellular matrix (ECM) in liver fibrosis. In a healthy liver, HSCs are quiescent and found in the subendothelial space of Disse and primarily store vitamin A. Upon exposure to inflammatory cytokines, mainly TGF-β, HSCs become activated, secreting ECM, primarily collagen type I [10], and releasing chemokines while upregulating inflammatory receptors.⬤Myofibroblasts are a major contributor to ECM deposition. These contractile cells are not present in the normal-functioning liver. Myofibroblasts have been identified to arise from liver-resident activated HSCs, portal fibroblasts, and bone marrow-derived fibrocytes as part of a wound repair process [10].⬤Liver sinusoidal endothelial cells (LSECs) are specialized cells lining the liver’s sinusoidal capillaries. Under normal conditions, LSECs form a permeable barrier with fenestrations, facilitating nutrient exchange, antigen presentation, leukocyte recruitment, and maintaining portal vascular tone. In fibrosis, LSECs lose their fenestrations, undergo capillarization, and contribute to early fibrosis, vasoconstriction, and angiogenesis [11]. Pathological angiogenesis driven by VEGF increases the stiffness of the liver microenvironment, further activating HSCs through mechanotransduction pathways. Additionally, neovessel formation leads to the production of transforming growth factor-beta (TGF-β), which also contributes to the activation of HSCs and promotes fibrosis [12].

Recent research has found specific markers associated with LSEC dedifferentiation, such as CD31, VEGFR2, LYVE-1, and stabilin-2, which can serve as potential biomarkers for monitoring fibrosis progression or regression. Stabilin-2, which has been found to reduce the concentration of low-density lipoproteins, the main molecule responsible for atherosclerosis and indirectly liver fibrosis, has been identified as a promising biomarker for liver fibrosis and a potential therapeutic target [13]. Additionally, the role of VEGFR2 in fibrosis and its therapeutic implications, such as the use of VEGF inhibitors and drugs that indirectly modulate the VEGFR receptor such as sorafenib to target LSECs, has been explored in the context of modulating angiogenesis and fibrosis [14,15]. Moreover, recent studies have emphasized the therapeutic potential of restoring LSEC fenestrations to counteract fibrosis, which could be a pivotal strategy in treating liver diseases [16].

The diagnostic and therapeutic potentials of these markers and approaches are distinct yet complementary. While diagnostic markers can track disease severity and treatment response, therapeutic strategies focus on reversing the underlying LSEC dysfunction. Further studies are warranted to explore the bidirectional relationship between LSEC marker expression and fibrosis regression, thereby advancing our understanding of their clinical utility in both diagnosis and treatment.

A key component in the pathogenesis of liver fibrosis is inflammation. Interactions between the various immune cells play a significant role in the pathogenesis of liver fibrosis. Inflammatory cells such as macrophages and neutrophils provide a source of TGF-β, reactive oxygen species (ROS), and lipid peroxidation [2]. CD4+ and CD8+ T cells have also been shown to modulate the process of fibrogenesis in the liver. Th17 cells, in particular, promote fibrosis via IL-17, which activates HSCs.

Metabolic factors such as non-alcoholic fatty liver disease (NAFLD) and metabolic syndrome have also been linked to fibrosis. Insulin resistance, lipotoxicity, and oxidative stress drive HSC activation and ECM deposition. Adipokines such as adiponectin and leptin influence liver fibrosis through their effects on HSCs and inflammation. Although hepatic macrophages (Kupffer cells) were initially thought to be major producers of TGF-β, evidence suggests that bone marrow-derived macrophages play a more significant part. Kupffer cells are involved in both fibrosis development and its regression, by producing matrix metalloproteinases (MMPs), particularly MMP9 and MMP12, which help degrade ECM during fibrosis resolution [17].

Extracellular matrix (ECM) remodeling is the hallmark of liver fibrosis. Disruption of matrix metalloproteinases can lead to ECM deposition and fibrosis. Natural killer cells have also been linked with regression of liver fibrosis by inducing apoptosis of hepatic stellate cells. Natural killer cells accomplish this through several mechanisms, including the degradation of cytotoxic molecules (granzyme B), activation of the caspase 3 and 8 pathways, and secretion of interferon-gamma [17].

Recent studies have also demonstrated the role of epigenetic modifications, such as DNA methylation, histone modification, and non-coding RNAs in the regulation of gene expression in fibrosis. Additionally, specific microRNAs (miRNAs) like miR-21, miR-29, and miR-122 have also been implicated in the pathogenesis of fibrogenic pathways [18]. The gut–liver axis has also been linked to fibrosis. Dysbiosis and increased intestinal permeability lead to the translocation of bacterial products like lipopolysaccharides (LPS) that exacerbate liver inflammation and fibrosis [19].

The primary mechanisms of morbidity due to the complications of cirrhosis are increased hepatic resistance and the development of hyperdynamic circulatory alterations. The increase in hepatic resistance is driven mainly by anatomical disruptions caused by fibrosis in advanced liver disease and by functional alterations such as endothelial dysfunction mentioned previously [20].

Portal resistance is regulated by an interplay of vasodilatory and vasoconstrictory mediators. When endothelial dysfunction ensues, there is a decreased intrahepatic production of nitric oxide (NO), the main vasodilator, and, simultaneously, there is increased intrahepatic production of vasoconstrictors such as thromboxane A2, norepinephrine, and endothelin-1 [20].

As an adaptive response to portal hypertension, the opposite effect of what happens in the intrahepatic circulation occurs in the splanchnic circulation. Nitric oxide is produced, causing widespread vasodilation. This determines an increase in the shunting of blood from the systemic circulation into the splanchnic circulation and, thus, an increase in portal circulatory inflow. At the same time, splanchnic vasodilation leads to a drop in systemic vascular resistance. To compensate for this and improve the overall effective circulating volume and tissue perfusion, there is an increase in heart rate and cardiac output. The development of hyperdynamic systemic and splanchnic circulation ensues, which aggravates portal pressure further. The drop in systemic vascular resistance causes a reduction in renal blood flow, activating the renin–angiotensin–aldosterone system that leads to increased production of the antidiuretic hormone. The end result is a positive feedback loop leading to increased portal pressure, development of collateral circulation, and increased hydrostatic pressure due to hyperdynamic circulation [20,21].

An important focus of the study is whether liver fibrosis is reversible (Figure 2). Several clinical and experimental studies have shown that liver fibrosis is reversible upon removal of the etiological source. Factors signaling this include decreased production of TGF-β and other inflammatory cytokines, apoptosis of HSCs and myofibroblasts, and dissolution of extracellular matrix proteins [22]. As a central player in liver fibrosis, the fate of hepatic stellate cells during regression has been a subject of significant research interest. Past studies suggested apoptosis as the primary mechanism by which these cells were eliminated, resulting in the reversion of fibrosis. However, evidence now shows that HSCs can revert to an inactivated phenotype. Although inactivated HSCs no longer participate in ECM deposition and fibrosis, they seem to retain a biological memory that facilitates a more rapid and extensive reactivation when re-exposed to liver injury [23]. This discovery has become a possible therapeutic target for antifibrotic treatment, and efforts are being made to block inactivated HSC reactivation. Whether cirrhosis can revert to an entirely normal liver architecture is still unknown.

### 1.4. Histopathology

Microscopic evaluation of cirrhotic liver samples varies depending on the etiology of the disease. In chronic viral hepatitis, the first sign seen is portal expansion. This is followed by periportal fibrosis and diffuse septal (bridging) fibrosis. In alcoholic liver disease and non-alcoholic fatty liver disease, a combination of fatty change with centrilobular, perivenular, and sinusoidal fibrosis is first seen. Later in the condition, fat may be absent or scanty as fibrosis increases. Biliary tract disease shows a “jigsaw” micronodular pattern and prominent “halos” due to feathery degeneration of periseptal hepatocytes [24,25].

Histopathological assessment of liver disease is primarily performed with hematoxylin and eosin stains (H&E). However, other stains are used to identify several features not seen on regular H&E stains. For example, an essential stain to identify type I collagen due to liver fibrosis is Masson’s trichrome stain. It is used to stage chronic liver diseases and helps to identify the pattern of fibrosis. Other examples of special stains include iron stain (Prussian blue reaction) when suspecting hemochromatosis and periodic acid–Schiff stain (PAS) in alpha-1 antitrypsin deficiency [26].

In the past, cirrhosis was classified into one of three different morphological categories [24]:⬤Macronodular: nodules > 3 mm, up to several cm in diameter. This morphology is primarily seen in cirrhosis due to chronic viral hepatitis and primary biliary cholangitis. Postmortem analysis showed liver sizes that were usually normal or reduced from scarring.⬤Micronodular: regular-sized, <3 mm in diameter nodules. Cirrhosis presenting with this morphological pattern is usually caused by alcoholism, biliary obstruction, venous outflow obstruction, and hemochromatosis. The size of this liver postmortem was often increased, especially with extensive fat deposition.⬤Mixed: when both patterns of nodules are present in almost an equal amount.

Although this classification was adopted for several years, it no longer carries the same importance due to having limitations. There are currently several laboratory markers that are more specific in identifying the etiology of cirrhosis. Moreover, morphological classification requires a biopsy sample obtained through invasive procedures. The most significant reason for abandoning this classification is that morphological features can change over time. An example of this is that micronodular cirrhosis usually converts to macronodular cirrhosis [27].

There is a functional specialization of liver tissue architecture known as metabolic zonation, which can be affected in diverse ways by liver diseases (Figure 3). There are three main zones [28]:⬤Zone I (periportal zone): affected first by viral hepatitis. It is also the most resistant zone to circulatory compromise due to receiving better oxygenation.⬤Zone II (intermediate zone): found to be a vital component of hepatocyte turnover and regeneration.⬤Zone III (pericentral vein or centrilobular zone): affected first by ischemia due to being the least oxygenated sector. It is also the zone most affected by metabolic toxins such as ethanol, carbon tetrachloride, and rifampin.

### 1.5. History and Physical

Very often chronic liver disease can be asymptomatic or may also present with nonspecific systemic symptoms like fatigue, weakness, anorexia, muscle cramps, and weight loss. More notorious symptoms manifest when complications of cirrhosis develop and lead to clinical decompensation. These symptoms stem from two main causes: portal hypertension and hepatic insufficiency. Once decompensation occurs, the signs and symptoms of cirrhosis are numerous and varied, affecting multiple organs and systems.

⬤Skin findings—jaundice, spider angiomas, palmar erythemas, easy bruisability [29].⬤Chest—gynecomastia, loss of chest hair and axillary hair distribution in males [30]⬤Abdomen—initial symptoms seen include anorexia, nausea and vomiting, and dull abdominal pain. However, the most characteristic manifestations are caused by portal hypertension. This includes hepatomegaly, splenomegaly, caput medusae, and ascites [31].○Spontaneous bacterial peritonitis can occur as a complication of cirrhosis and ascites. It is defined as the infection of ascitic fluid without an evident intraabdominal secondary source. Spontaneous bacterial peritonitis can present as fever, abdominal pain and tenderness, and altered mental status. Early recognition of spontaneous bacterial peritonitis is vital as the treatment window is narrow and can quickly decompensate into multisystem organ failure [32]. ○Esophageal varices can manifest as upper gastrointestinal bleeding. This is the most lethal complication of cirrhosis. It is estimated that around one-third of all patients with varices will develop variceal hemorrhages [33].⬤Renal—Hepatorenal syndrome constitutes a form of prerenal failure in the setting of severe liver fibrosis [34].⬤Genitourinary—Testicular atrophy, impotence, feminization, and infertility in men, amenorrhea in women [30].⬤Extremities—Clubbing, periostitis, peripheral edema [35,36].⬤Neurologic—Hepatic encephalopathy is usually seen in advanced liver disease. This manifests as confusion, decreased attention, memory problems, inappropriate behavior, mood changes, and disturbed sleep patterns [37].

## 2. Evaluation

### 2.1. Laboratory Findings

Since cirrhosis is frequently asymptomatic, laboratory findings may be the first indicator of hepatic damage.

Hematologic abnormalities: As the liver synthesizes most coagulation factors and anticoagulant proteins, coagulation disorders occur as hepatic function decreases. Prothrombin time is usually the first indicator to be prolonged (as factor VII has the shortest half-life). Anemia can occur due to several causes such as hypersplenism, folate deficiency, alcoholism, and gastrointestinal blood loss. Portal hypertension leads to splenomegaly which can cause sequestration of platelets and thrombocytopenia [38].

Liver function tests: Measurement of liver function tests is a standard screening method when initially suspecting cirrhosis. Tests evaluating the liver’s biosynthetic capacity, such as clotting factors and serum albumin, are significantly affected. Decreasing levels of albumin help can help grade the severity of cirrhosis [39].

The other components of the liver function test measure enzymes released during liver damage. In most forms of chronic liver disease, there is a rise in ALT and AST with an AST/ALT ratio < 1. However, normal levels do not rule out cirrhosis [40]. Alkaline phosphatase and gamma-glutamyl transferase levels can be elevated in cirrhosis, in cholestatic etiologies such as primary biliary cirrhosis and primary sclerosing cholangitis. Serum bilirubin levels may rise in many causes of cirrhosis [41].

Serum markers of fibrosis: Different serum markers have been developed to help evaluate the degree of liver fibrosis. These panels are subdivided into indirect or direct markers of fibrosis. Indirect markers measure a combination of serum parameters that evaluate liver function and predict the stage accordingly. On the other hand, direct markers identify changes in extracellular matrix deposition and metabolism.

Of the indirect markers, one of the most commonly used panels of tests is the Fibrotest and Fibroscore panels. When the FibroSure panel was compared to findings from a liver biopsy, which was used as the standard reference tool for the evaluation of liver fibrosis in a meta-analysis, it showed acceptable performance for the detection of cirrhosis. However, the test was not as accurate in detecting advanced fibrosis in NAFLD patients [42].

Of the direct markers, the FibroSpect II panel is often used. The FibroSpect II panel has been evaluated in comparison to findings seen on a liver biopsy, considered the gold standard test for the evaluation of liver fibrosis. The panel was able to successfully rule in significant fibrosis with a likelihood ratio of 2.6 [43].

The components of the panels are covered in Table 1.

While all three tools provide data on liver fibrosis, the choice between these tools may depend on availability, specific patient characteristics, and clinical context. These combination panels have demonstrated the capacity to discern between stages of no or mild fibrosis from those with severe fibrosis. However, a shortcoming identified is the lack of ability to differentiate between specific stages of fibrosis classification. Additionally, the percentage of test results labeled as indeterminate occurs in almost half the cases. For this reason, although serum markers have become an aid in predicting cirrhosis, there is still no test that is used as a reliable standard of care [44].

### 2.2. Imaging Studies

Ultrasound: This is the initial imaging study for hepatic fibrosis due to its noninvasive nature, high diagnostic accuracy, cost-effectiveness, and point-of-care application. It can detect liver echogenicity, surface nodularity, right lobe atrophy, caudate lobe hypertrophy, and other liver fibrosis indicators such as ascites, splenomegaly, and portal hypertension. Its diagnostic accuracy for cirrhosis ranges from 82–88% [45], but it is not specific and cannot reliably differentiate between hepatic fatty infiltration with liver fibrosis and cirrhosis [46]. It may also be affected by anatomical factors and interobserver variability.

CT/MRI: Both CT and MRI are not the first choice for evaluating cirrhosis due to radiation exposure, contrast agents, high costs, and lower patient tolerability. Many findings on CT/MRI can also be detected by ultrasound. However, they are helpful for screening hepatic vascular diseases and hepatocellular carcinoma with improved detection compared to ultrasound alone [47].

Transient elastography: Transient elastography, also known as FibroScan and approved by the FDA, is a noninvasive technique that measures the stiffness and elasticity of the liver using low-frequency vibratory vibrations produced by ultrasonography. Liver stiffness and wave propagation velocity are directly related. It is highly accurate in identifying advanced fibrosis and cirrhosis (F2 and greater) and can distinguish these from minimal or no fibrosis (F1 and F0). It thus detects early cirrhosis and has prognostic value [48].

Classification of fibrosis as per FibroScan is as follows:⬤F0: no cirrhosis⬤F1: portal fibrosis⬤F2: fibrosis with few septa⬤F3: fibrosis with many septa⬤F4: cirrhosis

Liver biopsy: This is the gold standard for diagnosing fibrosis and cirrhosis, but it is invasive, has potential complications, and is prone to sampling error. With patient history, lab results, and imaging, a biopsy is often unnecessary. It is only indicated when lab and imaging studies fail to provide a definitive diagnosis [49].

### 2.3. Prognosis

The prognosis of cirrhosis depends on the presence or development of complications. Patients with compensated cirrhosis are defined as those who do not experience complications. Patients with decompensated cirrhosis are those who develop major complications of cirrhosis from their cirrhosis. The differences in survival between the two are substantial. The median survival time in patients with compensated cirrhosis is >12 years, while in patients with decompensated cirrhosis is approximately two years. Patients with compensated cirrhosis mainly die after transitioning to decompensated cirrhosis at a rate of roughly 5–7% per year [50,51].

There are two main predictive models used to calculate survival in patients with end-stage liver disease:

**Child–Pugh score** [52]: Its variables include total bilirubin, serum albumin, INR, and evaluation of ascites and hepatic encephalopathy (Table 2).

Patients are then assigned to one of three classes:⬤Class A (5–6 points): well-compensated cirrhosis;⬤Class B (7–9 points): significant functional compromise;⬤Class C (10–15 points): decompensated cirrhosis.

**MELD score** [53]: This uses serum creatinine, total bilirubin, INR, serum sodium, and the use of hemodialysis at least twice or the use of continuous veno-venous hemodialysis (CVVHD) in the past week (Table 3).

Mortality is then calculated based on the score received:⬤Score ≤9: 1.9% mortality;⬤Score 10–19: 6.0% mortality;⬤Score 20–29: 19.6% mortality;⬤Score 30–39: 52.6% mortality;⬤Score ≥40: 71.3% mortality.

The MELD score is currently used for prioritizing patients awaiting liver transplantation [53].

### 2.4. Treatment/Management

The main goal for the management of cirrhosis is to halt the progression, and in some cases, reverse liver injury. This is achieved by addressing the specific underlying etiology of fibrosis. As examples:⬤Abstinence from alcohol in alcoholic cirrhosis;⬤Antiviral treatment achieving a sustained virologic response in patients with chronic viral hepatitis;⬤Weight loss and/or bariatric surgery in NAFLD;⬤Copper chelation for patients with Wilson disease;⬤Iron chelation and phlebotomy in hemochromatosis;⬤Ursodeoxycholic acid and obeticholic acid in primary biliary cholangitis;⬤Corticosteroids and immunosuppressants in autoimmune hepatitis;⬤Removal of the offending drug in drug-induced liver injury.

Additional goals of cirrhosis management include [22]:

Prevention of superimposed liver injury: All patients should receive vaccines for hepatitis A and B. Pneumococcal and yearly influenza vaccines are also recommended. Abstinence from alcohol, cigarette smoking, and hepatotoxic drugs should be stressed. Additionally, dose adjustments of a patient’s current medications may be indicated, considering the degree of hepatic compromise.

Maintaining adequate nutrition, weight, and lifestyle habits: Patients should be advised to have an antioxidant-rich diet and to keep a normal weight range. Insulin resistance, obesity, and metabolic syndrome have all been linked to fibrosis progression and increased mortality.

Prevention of complications: Patients with cirrhosis should be monitored for complications, and efforts should be made to decrease their occurrence. A few examples include:⬤Esophageal varices: All cirrhosis patients should be screened for varices with an upper endoscopy. The primary prophylactic therapy is with nonselective beta-blockers or endoscopic band ligation.⬤Hepatocellular carcinoma: Cirrhosis patients should undergo ultrasonography every six months for the surveillance of hepatocellular carcinoma.⬤Ascites: Patients should be educated on limiting dietary sodium consumption. When ascites develop, diuretic therapy should be initiated.⬤Infection: Patients with cirrhosis are predisposed to develop bacterial infections such as spontaneous bacterial peritonitis and urinary and respiratory tract infections. Primary prophylaxis with antibiotics has demonstrated survival benefits [50].⬤Hepatic encephalopathy: Avoidance of triggers and prevention of recurrence with lactulose therapy.

Liver transplantation is the final therapeutic option for patients with decompensated cirrhosis. Guidelines have been developed to help choose candidates who will benefit from transplantation. With the procedure, life expectancy is generally greatly improved. Estimates of 10-year patient survival rates surpass 70% in many indicators [54].

The exact moment when liver fibrosis becomes irreversible is still unknown. Reversion of liver fibrosis can be achieved, usually at the initial stages of cirrhosis [3]. In recent years, drugs have been approved that have been proven to reverse hepatic fibrosis.

### 2.5. Prospective Treatment Strategies [55]

Liver fibrosis is a progressive condition characterized by excessive extracellular matrix (ECM) deposition and hepatic stellate cell (HSC) activation, ultimately leading to cirrhosis and liver failure if left untreated. Despite the complexity of the underlying mechanisms, recent advancements in pharmacotherapy offer promising strategies for targeting key pathways involved in fibrosis. These emerging treatments focus on modulating nuclear receptors, lipid metabolism, chemokine signaling, and hormonal pathways to reduce liver injury and fibrosis. Drugs like Resmetirom (Figure 4), a thyroid hormone receptor beta agonist, and Lanifibranor, a pan-PPAR agonist, are being explored for their ability to promote lipid catabolism and inhibit fibrogenesis, thereby reducing HSC activation and ECM deposition. Similarly, FXR agonists like obeticholic acid (Figure 5) and lipotoxicity modulators like Aramchol target bile acid synthesis and lipid accumulation to alleviate hepatotoxicity and fibrosis. Additionally, chemokine receptor inhibitors, hormonal agonists such as GLP-1 receptor agonists, and FGF analogs are gaining attention for their potential to modulate inflammatory signals and inhibit HSC activation (Table 4).

### 2.6. Deterrence and Patient Education

Actively encouraging patients to engage in lifestyle modifications is a vital factor in preventing the progression of cirrhosis. While many of the cases of cirrhosis are irreversible, several measures can be taken to avoid the development of complications, including:⬤Abstinence from alcohol and smoking;⬤Avoidance of known hepatotoxic drugs (acetaminophen, amoxicillin/clavulanate);⬤Vaccination against hepatitis A and B, influenza, pneumococcus;⬤Reducing consumption of sodium and maintaining a balanced and high-fiber diet to ensure an adequate lipid profile;⬤Ultrasound screening for hepatocellular carcinoma every six months;⬤Screening for esophageal varices with upper endoscopy.

### 2.7. Pearls and Other Issues

Esophageal varices constitute a significant cause of morbidity and mortality in patients with cirrhosis. Esophageal varices can be found in approximately 50% of patients with cirrhosis, and variceal bleeding occurs at a rate of roughly 10–15% per year, supporting the need for screening for varices with upper endoscopy. In patients with compensated cirrhosis and esophageal varices, primary prevention of variceal hemorrhage includes the use of nonselective beta-blockers (propranolol, nadolol), carvedilol, or endoscopic variceal ligation. Patients on primary prophylaxis with beta-blockers do not require regular follow-up endoscopy [33].

### 2.8. Enhancing Healthcare Team Outcomes

Cirrhosis can lead to life-threatening complications and has an impact on the quality of life of patients. Therefore, it requires an interdisciplinary team of professionals, including gastroenterologists/hepatologists, internists, primary care providers, nutritionists, transplant surgeons, pharmacists, and nurse practitioners.

The primary care provider can investigate early suspicion of cirrhosis, by obtaining initial laboratory and imaging studies. Once a diagnosis of cirrhosis is established, coordination between the teams is essential as the disease must be closely monitored to prevent complications and further progression of liver fibrosis. Healthcare professionals should also regularly follow up on patients who are eligible for receiving liver transplantation.

## 3. Conclusions

Liver fibrosis has been a public health concern for a long time. Recent advancements in the understanding of pathogenesis have allowed for the potential development of pharmaceutical interventions to target the various mediators of fibrosis. However, the mainstay of treatment is still early detection of disease and prevention of development of complications. Clinicians should still focus on eradicating the underlying causes of cirrhosis as this has been proven to slow down the progression of disease. The approval of a drug by the FDA for NASH cirrhosis is encouraging. While the long-term benefits of this drug are currently unknown, this can be a part of treatment protocols in patients who qualify for treatment. However, the current standard of care still stresses lifestyle modifications and symptom modification, which should be continued.

## Figures and Tables

**Figure 1 biomedicines-12-02229-f001:**
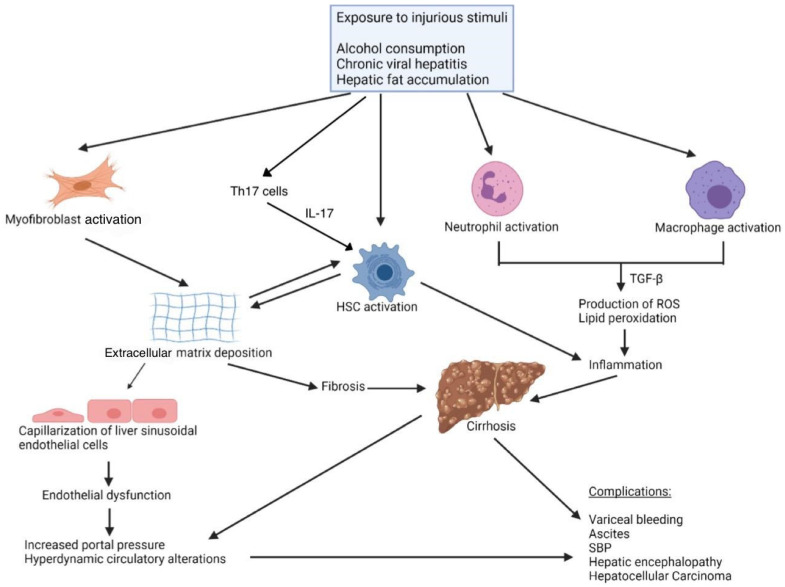
Pathogenesis of fibrosis. The pathogenesis of fibrosis involves the activation of various cell types in response to injurious stimuli and the subsequent interplay between them. Some ways this is accomplished are cytokine release leading to HSC activation and extracellular matrix deposition, and neutrophil and macrophage activation leading to inflammation via cytokine release.

**Figure 2 biomedicines-12-02229-f002:**
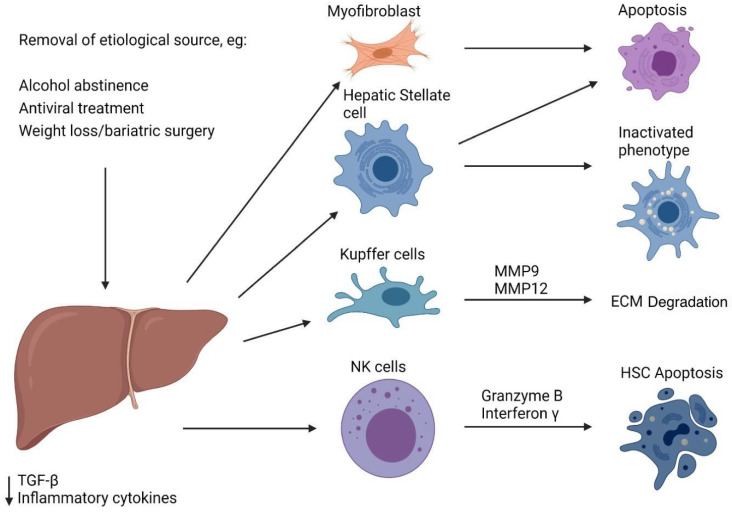
Target checkpoints in fibrosis progression. This figure details various checkpoints that can be intercepted to halt progression of liver fibrosis. Possible interceptions include decreasing production of TGF-beta and other inflammatory cytokines, inducing the apoptosis of activated HSCs and the dissolution of formed extracellular matrix proteins.

**Figure 3 biomedicines-12-02229-f003:**
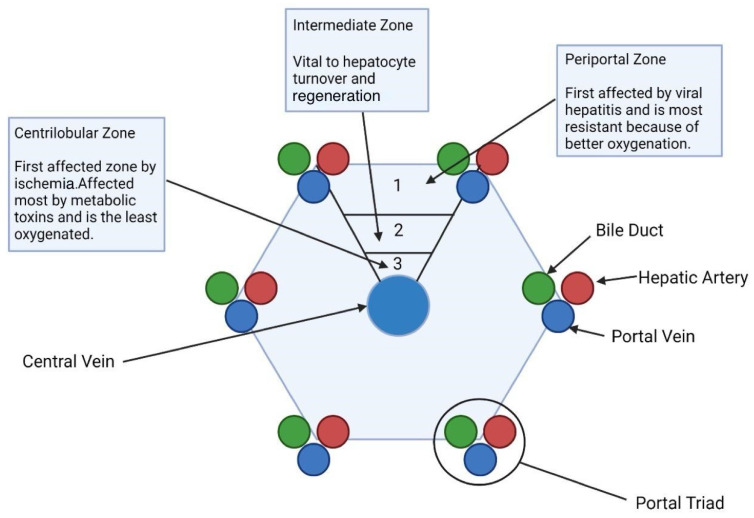
Liver zonation and susceptibility. This figure describes liver zonation, detailing functions and disease susceptibility particular to the centrilobular, intermediate, and periportal zones.

**Figure 4 biomedicines-12-02229-f004:**
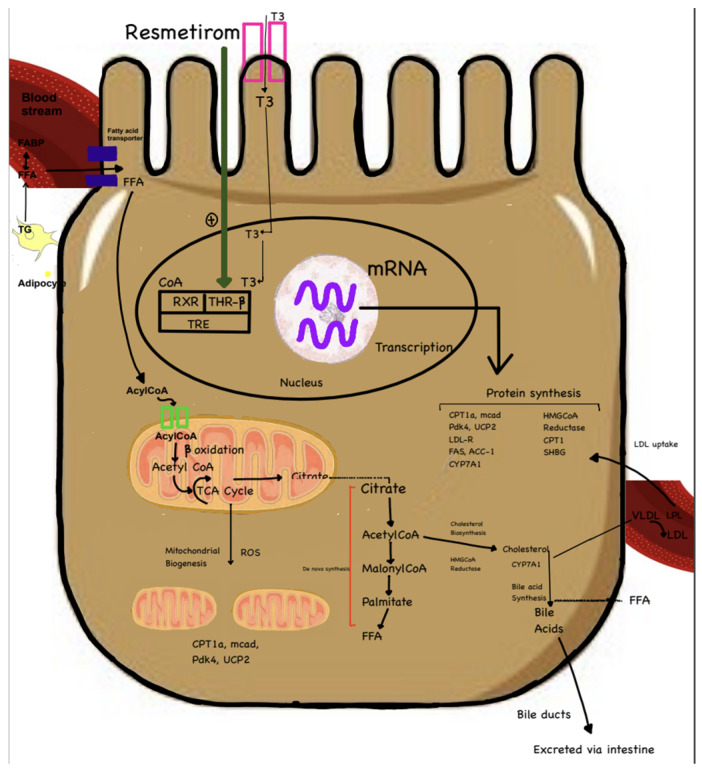
Mechanism of action of Resmetirom. Resmetirom is a partial THR-beta agonist. It promotes beta-fatty acid oxidation in the liver, reducing fat content in the liver. The activation of THR-beta reduces levels of intrahepatic triglycerides. Figure created by Rukam Mahawa MBBS, Sreeja Moolamalla MBBS.

**Figure 5 biomedicines-12-02229-f005:**
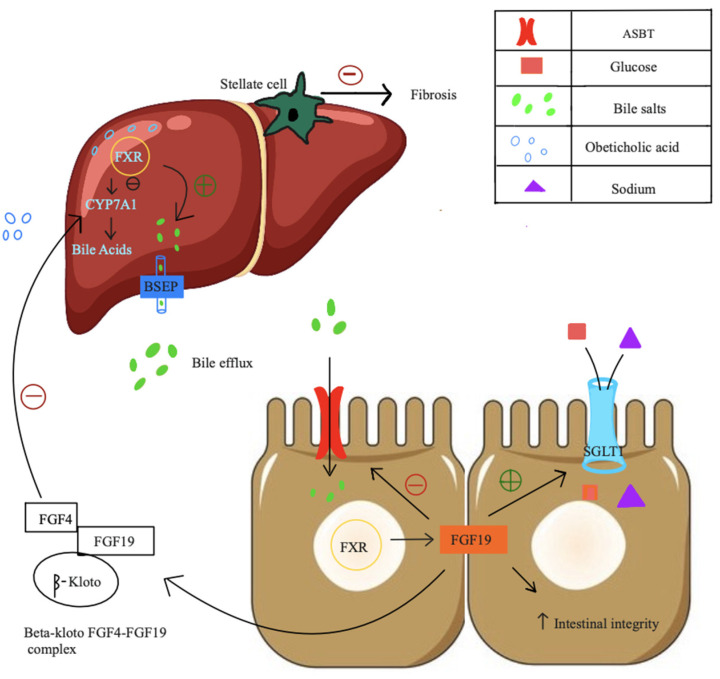
Mechanism of action of obeticholic acid. Obeticholic acid is a Farsenoid X receptor (FXR) agonist that, once bound, triggers a cascade leading to reduced bile production in the liver. It also facilitates bile removal by inducing FGF19 release and inhibiting CYP7A1 gene expression. Figure created by Rukam Mahawa, MBBS, and Sreeja Moolamalla, MBBS.

**Table 1 biomedicines-12-02229-t001:** Direct and Indirect markers used in the evaluation of liver fibrosis.

Direct Markers	Indirect Markers	
FibroSpect II Panel	Fibrotest/Fibrosure Panel	FibroScore Panel
Serum hyaluronic acid	Age of patient	Urokinase Plasminogen Activator
Tissue inhibitor of metalloproteinase-1 (TIMP-1)	Sex of patient	Matrix metalloproteinase-9 (MMP-9)
Alpha-2 macroglobulin	Alpha-2 macroglobulin	Beta-2 microglobulin
	Haptoglobin	
	Gamma globulin	
	Apolipoprotein A1	
	Gamma glutamyl transferase	
	Total Bilirubin	

**Table 2 biomedicines-12-02229-t002:** Components of the Child-Pugh Scoring system.

Parameter.	1 Point	2 Points	3 Points
Total bilirubin	<2 mg/dL	2–3 mg/dL	>3 mg/dL
Serum albumin	>3.5 g/dL	2.8–3.5 g/dL	<2.8 g/dL
INR	<1.7	1.7–2.2	>2.2
Ascites	Absent	Slight	Moderate
Encephalopathy	None	Grade 1–2	Grade 3–4

**Table 3 biomedicines-12-02229-t003:** Components of the MELD scoring system.

Parameter	Input
Creatinine	(mg/dL)
Total bilirubin	(mg/dL)
INR	
Serum sodium	mEq/L
Hemodialysis at least twice or CVVHD in the past week	Yes/No
Result calculated:	

**Table 4 biomedicines-12-02229-t004:** Current drugs in development/approved for use for liver fibrosis.

Drug Target	Drug	Mechanism of Action	Comments
Nuclear Receptor Agonists			
1.Thyroid hormone receptor beta agonist	Resmetirom	Promotes intrahepatic lipid catabolism and bile acid remodeling leading to decreased hepatic lipid accumulation and reduction in HSC activation	Approved by the FDA for use in 2024
2.Pan-PPAR agonist	Lanifibranor	Improved lipid metabolism and inhibition of fibrogenesis leading to reduced HSC activation and decreased ECM deposition	
3.FXR agonist	Obeticholic acid	Decreases bile acid synthesis and promotes bile acid export, reducing hepatotoxicity. Also inhibits HSC activation and modulates TGF- Beta signaling pathway, decreasing fibrosis	Currently used for primary biliary cholangitis. Its potential therapeutic benefits in liver fibrosis are being evaluated as of now
Lipotoxicity modulators	Aramchol	Downregulates hepatic stearoyl-CoA desaturase 1 and modulates AMPK/mTORC1 pathways to reduce lipid accumulation and fibrosis	
Acetyl-CoA carboxylase	Decreases de novo lipogenesis and directly impair activation, thereby mitigating hepatic fibrosis	
Chemokine receptor inhibitors	Galectin-3 inhibitors, anti-platelet drugs	Reduction of immune cell activation and recruitment	
Hormonal agonists	GLP1-RASemaglutide, Tirzepatide	Modulation of extrahepatic inflammatory signals from the adipose tissue and gut	
FGF-19 and FGF-21 analogs	Reduces bile acid synthesis and anti-fibrogenic gene expression, further inhibiting HSC activation and promoting apoptosis, thereby attenuating hepatic fibrosis through multiple pathways

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
