# Peer review of "Liver Fibrosis Leading to Cirrhosis: Basic Mechanisms and Clinical Perspectives"

_biomedicines, 2024, doi:10.3390/biomedicines12102229_

Round 1
Reviewer 1 Report
Comments and Suggestions for Authors
Liver fibrosis is a dynamic pathological condition characterized by excessive extracellular matrix remodeling, often progressing to cirrhosis with iterative liver damage with no effective treatments. The interplay of various, parenchymal and nonparenchymal cells, inflammatory processes, and metabolic factors contributes to both the progression and regression of bidirectional fibrogenesis. Hepatologists worldwide are observing an emerging posttreatment patient pool exhibiting a trend that even cirrhosis appears to be reversible after the iterative injury (such as viral infection) is eradicated or suppressed. As invasive liver biopsies are increasingly declined by patients, research into the mechanisms driving bidirectional fibrogenesis may enable the development of models for the non-invasive assessment of fibrosis kinetics, utilizing markers like liver stiffness measurements from baseline to post-treatment follow-up. The current report may aid in understanding bidirectional the liver disease severity and the treatment responses, and elucidating these liver fibrosis surrogate biomarkers in terms of the mechanistic gaps when applied to risk stratification after applying liver therapeutics.
Concerns:
- The section addressing LSECs does present a biologically plausible explanation of their role in liver fibrosis. It addresses the fundamental LSEC capillarization, loss of fenestrations, and their contribution to fibrosis, vasoconstriction, and angiogenesis. However, the section could benefit from the concise yet explicit inclusion of more specific published citations and ongoing research relevant to LSECs in the context of monitoring bidirectional disease severity and treatment response or fibrosis regression. Here are some recommendations:
1. LSEC-relevant markers: address specific human or animal markers associated with LSEC dedifferentiation or capillarization (e.g., CD31, VEGFR2, LYVE-1, stabilin-2, etc.) and how their expressions could be utilized as potential biomarkers in monitoring fibrosis progression or regression. 2. LSEC-targeted therapies: Concisely discuss emerging, potential therapeutic approaches targeting LSECs to modulate fibrosis (e.g., anti-angiogenic agents, VEGF inhibitors, or therapies aimed at restoring LSEC fenestrations).
- Concisely distinguish between the diagnostic and therapeutic potential of the LSEC markers addressed, while clarifying their cellular localization in relation to the central dogma.
Comments on the Quality of English LanguageNone.
Author Response
Feedback Reviewer 1: LSECs and their markers
Author changes: Added more LSECs-specific markers regarding their potential use in therapeutics of liver fibrosis along with relevant studies that have been done with regard to these markers on from lines 94-110 in the file.
Reviewer 2 Report
Comments and Suggestions for Authors
The presented literature review discusses the current evidence on the mechanisms of fibrosis and liver fibrosis. Current views on the role of stellate cells, migrating and resident macrophages are presented. Clinical data and current therapies are reviewed. The article is well written and illustrated with diagrams.
A small remark: the authors need to provide the names of figures, currently they are missing.
Author Response
Feedback reviewer 2: Image names and descriptions
Author changes: Titles and descriptions have been added to figures with relevant acknowledgments. Changes have been made in lines 165-169, 185-188, 234-236, 438-441, 468-471
Reviewer 3 Report
Comments and Suggestions for Authors
This is a comprehensive review of liver fibrosis, covering a wide range of topics from basic medicine to current diagnosis and treatment. The article provides clear and concise explanations of very important information and is considered useful for busy clinicians to organise their knowledge. However, there is a tendency to oversimplify the explanations. For example, section 2.5, Prospective Treatment Strategies, contains only tables and figures and no written explanations, which may not provide the reader with sufficient understanding. I would like to request that the manuscript be revised to describe the reader's perspective.
Comments on the Quality of English LanguageAuthors should be required to check themselves. For example, in Figure 2, 'degradation' is misspelled as 'degradation'.
Author Response
Feedback reviewer 3: Introduction and explanation for section 2.5 Prospective treatment modalities
Author changes: Added an introductory paragraph that transitions into a prospective treatment strategies table in lines 422-436 to aid in reader understanding of section 2.5.